# Hippo Pathway Dysregulation in Thymic Epithelial Tumors (TETs): Associations with Clinicopathological Features and Patients’ Prognosis

**DOI:** 10.3390/ijms26135938

**Published:** 2025-06-20

**Authors:** Lisa Elm, Nadja Gerlitz, Anke Hochholzer, Thomas Papadopoulos, Georgia Levidou

**Affiliations:** Department of Pathology, Nuremberg Clinic, Paracelsus Medical University, 90419 Nuremberg, Germany; lisa.elm@klinikum-nuernberg.de (L.E.);

**Keywords:** thymic epithelial tumors, TETs, hippo signaling pathway, immunohistochemistry

## Abstract

Thymic epithelial tumors (TETs) display heterogeneous histology and often unpredictable clinical behavior. The Hippo signaling pathway has been implicated in tumorigenesis, but its role in TETs remains poorly characterized. We performed the first comprehensive immunohistochemical analysis of core and upstream Hippo pathway components—YAP1, active YAP (AYAP), TAZ, LATS1, MOB1A, MST1, SAV1, and TEAD4—in 77 TETs. Associations with clinicopathological parameters and survival were explored. We observed widespread expression of Hippo components in TETs with significant associations among molecules and differences in subcellular localization and expression in normal tissue. Early stage TETs showed higher nuclear YAP1 (*p* = 0.032) and AYAP (*p* = 0.007), while cytoplasmic MST1 (*p* = 0.002), LATS1 (*p* = 0.007), MOB1A (*p* = 0.033) and TEAD4 (*p* < 0.001) correlated with advanced stage. Cytoplasmic MST1 (*p* = 0.014), LATS1 (*p* < 0.001) and TEAD4 (*p* = 0.005) were associated with histological aggressiveness. Cytoplasmic TEAD4 overexpression was associated with poorer overall survival (log-rank, <70% versus ≥70%, *p* = 0.003). Our findings provide novel insights into the differential regulation and compartmentalization of Hippo components in TETs. While indolent tumors show features that are consistent with partial Hippo inactivation, more aggressive phenotypes exhibit reduced nuclear YAP/TAZ and altered TEAD4 compartmentalization, suggesting a context-dependent Hippo signaling state. Cytoplasmic TEAD4 emerges as a potential adverse prognosticator, indicating involvement in non-canonical or Hippo-independent mechanisms.

## 1. Introduction

The thymus, situated in the anterior mediastinum, is essential for immunological development, especially in the maturation and selection of T-cells. This organ comprises different cortical and medullary areas, each inhabited by specialized epithelial cells [1,2,3]. Malignant transformation of thymic epithelial cells leads to the growth of thymic epithelial tumors (TETs), a rare yet clinically relevant category of neoplasms exhibiting epithelial traits [1,4,5,6]. Although infrequent, constituting merely a small fraction (0.2% to 1.5%) of all human malignancies, TETs are the predominant neoplasms located in the anterior mediastinum [7].

From a pathological perspective, the World Health Organization (WHO) categorizes TETs into six specific subtypes (A, AB, B1, B2, B3, and thymic carcinoma (TC)), according to the lymphocyte-to-epithelial cell ratio and the tumor’s physical characteristics [7,8,9]. However, histological features alone cannot consistently predict tumor activity, as benign-appearing thymomas may have aggressive, invasive traits [7,10]. Tumor invasiveness is a more precise prognostic factor, as 15-year survival drops from 47% in non-invasive to 12.5% in invasive TETs. [7,10,11,12]. The Masaoka-Koga staging system continues to be the established approach for evaluating tumor invasion [4,6,10,11,12]. Distinguishing indolent from aggressive TETs remains difficult, highlighting the value of molecular markers for diagnosis and prognosis [13,14].

The Hippo pathway has received attention among the signaling pathways involved in cancer due to its regulation of cell proliferation, apoptosis, and organ size [1,15,16,17]. Moreover, it plays a critical role in the development and homeostasis of the thymus [13,14]. The Hippo signaling pathway is an evolutionarily conserved kinase cascade that modulates cell development and inhibits cancer (Figure 1) [1,15].

This pathway mostly encompasses the kinases MST1/2 (Mammalian STE20-like kinases) and LATS1/2 (large tumor suppressor kinases). Upon activation of MST1/2, it phosphorylates LATS1/2, which then phosphorylates the transcriptional co-activators TAZ (transcriptional co-activator with PDZ-binding motif) and YAP (Yes-associated protein) [1,15,17,18]. The phosphorylation leads to the sequestration and degradation of YAP and TAZ in the cytoplasm, preventing their nuclear translocation and consequent activation of gene expression [1,15,17,18]. The Hippo pathway is further controlled by adaptor proteins such as MOB1 (Mps one binder 1) and SAV1 (Salvador homolog 1). MOB1 enhances the activation of LATS1/2 by stabilizing MST1/2, while SAV1 acts as a scaffold protein, facilitating MST1/2 kinase activity. In the absence of Hippo pathway activity, unphosphorylated YAP and TAZ translocate to the nucleus, where they interact with transcription factors like TEAD4 (TEA domain transcription factor 4) to augment the expression of genes related to cell proliferation, survival, and tissue growth [1,13,15,17,18,19].

Beyond its role in tumor progression, YAP/TEAD-mediated signaling has recently been associated with resistance to targeted therapies in other tumor entities, such as KRAS- or EGFR-mutated cancers. There, YAP/TAZ activation enables transcriptional programs that bypass receptor tyrosine kinase (RTK) inhibition, thereby diminishing the efficacy of small-molecule inhibitors [20,21]. These findings emphasize the broader oncogenic relevance of Hippo signaling components and support the investigation of their expression and activity in rare tumors such as TETs. In parallel, several small-molecule inhibitors designed to disrupt the YAP–TEAD interaction are currently under clinical investigation [22], further underscoring the translational relevance of this pathway.

The deregulation of the Hippo pathway is linked to various malignancies, and the uncontrolled activation of YAP/TAZ is proposed to promote abnormal cellular proliferation and survival in numerous tumors, facilitating tumor progression [1,18]. Although advances have been made in understanding TET biology, the Hippo pathway’s role and mechanisms of dysregulation remain poorly understood and insufficiently studied [1]. Notably, a recent study by Palamaris et al. (2023) has laid the groundwork by investigating the expression of key Hippo pathway components in TETs, highlighting their potential as molecular markers [1].

This study extends previous analyses by investigating both core components (YAP1, TAZ, LATS1, and TEAD4) and upstream regulators (MOB1A, MST1, and SAV1) of the Hippo pathway across different TET subtypes and their clinical associations. Additionally, we analyze expression correlations between these components to identify possible functional interactions and gain deeper insight into Hippo pathway regulation in TETs.

The inclusion of these additional components is vital for gaining a deeper understanding of the pathway’s activity and regulation in TETs. Additionally, we analyze the expression of active YAP, which is particularly important for assessing the functional state of YAP and its potential role in promoting cell proliferation and tumor progression. By examining these relationships, we aim to evaluate molecular markers that could potentially improve risk stratification, support personalized therapy, and enhance outcomes in these rare, challenging tumors.

## 2. Results

To provide a concise overview of the major findings, Table 1 summarizes the expression patterns and localization of all investigated Hippo pathway components across the analyzed TET subtypes and stages. This summary serves as a guide to the more detailed descriptions in the subsequent sections. Our results are presented in a sequence according to the position of the investigated molecules within the signaling pathway, from upstream to downstream.

### 2.1. Immunohistochemical Expression of MST1 in TETs

In general, approximately 50% of thymocytes exhibited positive cytoplasmic staining in normal thymic tissue, while all other cell populations remained negative. Positive staining was observed in macrophages and occasionally in lymphocytes (Figure 2A).

MST1 expression in TETs was cytoplasmic and present in the vast majority of the cases (74/77, 96.1%), with a median value of 10% (range 0–100%). Only one of the three negative cases displayed nuclear immunopositivity. The vast majority of the cases displayed moderate staining intensity (71.4%), whereas 9.1% showed strong and 19.4% mild staining intensity. In six cases we also found a positive immunoreactivity in lymphocytes.

MST1 cytoplasmic expression was higher in TCs compared to thymomas (Mann–Whitney U test, *p* = 0.014, Figure 2D) and in the advanced tumor stage (Mann–Whitney U test, *p* = 0.002, Figure 2B,C,E). Moreover, when we compared the expression levels between B3 thymomas and TCs with the remaining histological types, B3 thymomas and TCs displayed a significantly higher MST1 expression (Mann–Whitney U test, *p* = 0.032, Figure 2D). There was no significant association with tumor size (Spearman’s correlation coefficient, *p* = 0.152), patients’ overall survival (log-rank test, >10% versus ≥10%, *p* = 0.99) or the remaining clinicopathological parameters as presented in Table A1.

### 2.2. Immunohistochemical Expression of SAV1 in TETs

In normal thymic tissue, SAV1 expression was cytoplasmic strongly positive. Likewise, both adipose and connective tissues exhibited robust positivity (Figure 3A).

SAV1 immunoreactivity in TETs was cytoplasmic and was observed in all the cases (77/77, 100%), showing an extensive expression in the majority of the cases (median value 100%, range 80–100%, Figure 3B,C). Overall, 74% of the cases displayed a strong staining intensity, 24.7% a moderate and only one case (1.3%) a mild staining intensity. In seventeen cases (22.1%), a positive staining was also observed in the lymphocytes.

In all but one of the B3/TC cases (96.4%) we observed an expression of SAV1 in all tumor cells, whereas in 24.5% of the rest WHO histological types there were also negative tumor cells within the tumor area (Fisher’s exact test, *p* = 0.025).The same applied to cases with advanced Masaoka-Koga stage, which all displayed a positive expression of SAV1 in all tumor cells, whereas in 23.4% of the stage I/II cases we found also negative tumor cells within the tumor (Fisher’s exact test, I/II versus III/IV, *p* = 0.027). There was no significant association with tumor size (Spearman’s correlation coefficient, *p* = 0.237), patients’ overall survival (log-rank test, <100% versus 100%, *p* = 0.479) or the remaining clinicopathological parameters, as presented in Table A2.

### 2.3. Immunohistochemical Expression of LATS1 in TETs

In normal thymic tissue, LATS1 exhibited weak cytoplasmic positivity, thymocytes showed nuclear positivity, and lymphocytes were negative (Figure 4A).

LATS1 expression in TETs was cytoplasmic and present in all cases (77/77), with a median value of 90% (range 10–100%). In fourteen cases (18%), we observed an intense staining, in 26 (33.8%) a moderate, in 29 (37.5%) mild and in 8 (10.4%) a very light stain in terms of intensity. In 27 cases (35.1%), we also had a positive stain in the lymphocytes. Positive LATS1 lymphocytes were mostly observed in B1-B2 thymomas as well as in AB thymomas (Fisher’s exact test, *p* < 0.001).

B3 Thymomas and TCs displayed significantly higher levels of LATS1 expression compared to the remaining WHO histological types (Mann–Whitney U test, *p* < 0.001, Figure 4B–D). In particular, TCs seemed to have the highest levels of expression when compared to thymomas, as displayed in Figure 4D (Mann–Whitney U test, *p* < 0.001).The same applied to cases with advanced Masaoka-Koga stage (III/IV), which displayed higher levels of LATS1 expression (Mann–Whitney U test, *p* = 0.007, Figure 4E). LATS1 expression was not correlated with the patients’ overall survival (log-rank test, *p* = 0.838) and the remaining clinicopathological parameters, as presented in Table A3.

### 2.4. Immunohistochemical Expression of MOB1A in TETs

Normal thymic tissue demonstrated diffuse cytoplasmic immunoreactivity in both thymic epithelial cells and thymocytes, while connective and adipose tissue, as well as vascular structures serving as internal negative controls, remained entirely unstained (Figure 5A). In general, the staining intensity observed in non-neoplastic regions was markedly lower than that observed in tumor tissue.

MOB1A expression in TETs was cytoplasmic and was present in all the examined cases (76/76, 100%), with a median value of 100% (range 10–100%, Figure 5B,C). The majority of the cases (47/76, 59.2%) have displayed an intense stain, 26.3% a moderate and 13.5% a mild staining intensity. In 30 cases, we also observed a positive staining in the lymphocytes (39.5%). Positive MOB1A lymphocytes were mostly observed in B1-B3 thymomas as well as in AB thymomas (Fisher’s exact test, *p* = 0.014).

B3 thymomas and TCs tended to have higher levels of MOB1A expression (Mann–Whitney U test *p* = 0.063, Figure 5D), but this relationship was of marginal significance. Additionally, advanced Masaoka-Koga stage cases showed higher levels of MOB1A expression (Mann–Whitney U test, I/II versus III/IV, *p* = 0.033, Figure 5E). MOB1A expression was not correlated with the patient’s overall survival (log-rank test, >100% versus 100%, *p* = 0.592), and the remaining clinicopathological parameters, as presented in Table A4.

### 2.5. Immunohistochemical Expression of YAP1 in TETs

Immunohistochemical analysis of YAP1 revealed stronger staining in normal thymic tissue compared to thymomas. In normal tissue, thymocytes showed mild to moderate nuclear and cytoplasmic positivity, while lymphocytes were negative (Figure 6A). Endothelial cells (vessels) displayed both nuclear and cytoplasmic YAP1 expression.

YAP1 expression in TETs was both nuclear and cytoplasmic. Nuclear YAP1 expression was observed in 90.9% (70/77, median value 40%) and cytoplasmic in 63.4% (44/77, median value 15%) of the cases. Most of the cases displaying nuclear immunoreactivity had either a moderate (34/70) or a mild (29/70) staining intensity, whereas most of the cases displaying cytoplasmic immunoreactivity had a moderate staining intensity (31/44). Nuclear and cytoplasmic YAP1 expression were positively correlated (Spearman’s correlation coefficient, R = 0.32, *p* = 0.004).

Nuclear YAP1 expression was higher in thymomas compared to TCs (Mann–Whitney U test, *p* = 0.001, Figure 6B–D) and in early tumor stage (Mann–Whitney U test, I/II versus III/IV, *p* = 0.023). Moreover, when we compared the expression levels between B3 thymomas and TCs with the remaining histological types, B3 thymomas and TCs displayed a significantly lower nuclear YAP1 expression (Mann–Whitney U test, *p* = 0.010, Figure 6D). Advanced tumor stage cases displayed higher levels of cytoplasmic YAP1 expression (Mann–Whitney U test, I versus II/III/IV, *p* = 0.032, Figure 6E). There was no significant association between either nuclear or cytoplasmic YAP1 expression and patients’ overall survival (log-rank test, <40% versus ≥40%, *p* = 0.323 for nuclear expression and <15% versus ≥15%, *p* = 0.468 for cytoplasmic expression) or the remaining clinicopathological parameters, as presented in Table A5 and Table A6.

### 2.6. Immunohistochemical Expression of Active YAP (AYAP) in TETs

In normal thymic tissue, thymocytes exhibited moderate nuclear positivity, predominantly with a strong staining intensity (Figure 7A).

AYAP immunohistochemical expression in TETS was both nuclear and cytoplasmic. Nuclear immunopositivity was observed in 72 cases (72/77, 93.5%, median value 40%) and cytoplasmic in 58 cases (58/77, 75.3%, median value 30%). Five cases did not display either nuclear or cytoplasmic AYAP expression, 58 showed both nuclear and cytoplasmic immunoreactivity, and the remaining 14 cases showed only nuclear expression. The vast majority of the examined cases showed a staining of mild intensity in both immunolocalizations, followed by moderate staining intensity, with only 5 cases having a strong nuclear and only 1 a strong cytoplasmic immunoreactivity. Nuclear and cytoplasmic AYAP expression were positively correlated (Spearman’s correlation coefficient, R = 0.58, *p* < 0.001).

Nuclear AYAP expression was higher in thymomas compared to TCs (Mann–Whitney U test, *p* < 0.001, Figure 7B–D) and in early tumor stage (Mann–Whitney U test, I/II versus III/IV, *p* = 0.007, Figure 7E). Moreover, type A/AB thymomas displayed the highest levels of nuclear AYAP expression, followed by type B thymomas, whereas TCs had the lowest levels of nuclear AYAP expression (Kruskal–Wallis ANOVA, *p* = 0.001, Figure 7D). Type A/AB thymomas also displayed the highest levels of cytoplasmic AYAP expression, followed by TC and B1-B3 thymomas (Kruskal–Wallis ANOVA, *p* = 0.011, Figure 7F). Both nuclear and cytoplasmic AYAP expression were not correlated with patients’ overall survival (log-rank test, <40% versus ≥40%, *p* = 0.296 for nuclear expression and <30% versus ≥30%, *p* = 0.614 for cytoplasmic expression, Table A7). The associations of nuclear or cytoplasmic AYAP with the remaining clinicopathological parameters are presented in Table A7.

### 2.7. Immunohistochemical Expression of TAZ in TETs

TAZ generally shows low cytoplasmic expression in normal thymic tissues, with sporadic mild nuclear positivity observed (Figure 8A). Thymocytes occasionally exhibited nuclear positivity, while endothelial cells showed strong positivity.

TAZ expression in TETs was both nuclear and cytoplasmic. Nuclear expression was observed in 68.8% of the cases (53/77) with a median value of 10% (range 0–90) and cytoplasmic in 51.9% of the cases (40/77) with a median value of 2% (range 0–90%). Most of the positive cases displayed a rather moderate staining intensity. In total, 28 cases displayed both nuclear and cytoplasmic immunopositivity, 25 only nuclear, 12 only cytoplasmic and 12 did not show any nuclear or cytoplasmic immunopositivity. There was no association between nuclear and cytoplasmic TAZ expression (Spearman’s correlation coefficient, *p* = 0.814).

There was a significant difference in nuclear TAZ expression among different categories of TETs, thymomas B1-B3 showing the lowest levels of expression, followed by TCs, whereas TETs containing an A component (A or AB) had the highest levels of nuclear TAZ expression (Kruskal–Wallis ANOVA, *p* = 0.004, Figure 8B–D). On the other hand, cytoplasmic TAZ expression was significantly lower in B3/TC compared to the rest WHO histological types (Mann–Whitney U test, *p* = 0.004, Figure 8D). There was no significant correlation with the Masaoka-Koga stage (Mann–Whitney U test, I/II versus III/IV, *p* = 0.182 for nuclear expression and *p* = 0.281 for cytoplasmic expression), or with patient’s overall survival (log-rank test, <10% versus ≥10%, *p* = 0.951 for nuclear expression, <2% versus ≥2% *p* = 0.218 for cytoplasmic expression, or the remaining parameters, as presented in Table A8 and Table A9.

### 2.8. Immunohistochemical Expression of TEAD4 in TETs

In normal thymic tissue, nuclear positivity for TEAD4 was observed, while cytoplasmic staining was weakly positive. Lymphocytes remained negative (Figure 9A).

TEAD4 expression in TETs was both nuclear and cytoplasmic. Nuclear immunopositivity was observed in 49 cases (49/77, 63.6%, median value 15%) and cytoplasmic in 73 cases (73/77, 94.8%, median value 70%). Overall, 2 cases did not display either nuclear or cytoplasmic TEAD4 expression, 47 showed both nuclear and cytoplasmic immunoreactivity, 26 displayed only cytoplasmic expression and the remaining 2 cases showed only nuclear expression. The majority of the examined cases showed a nuclear staining of mild intensity and a cytoplasmic staining of moderate intensity, with only 1 case having a strong nuclear and only 6 a strong cytoplasmic immunoreactivity. Nuclear and cytoplasmic TEAD4 expression were positively correlated (Spearman’s correlation coefficient, R = 0.31, *p* = 0.007). In four cases we also observed a positive immunoreactivity in the lymphocytes.

Cytoplasmic TEAD4 expression was higher in TCs compared to thymomas (Mann–Whitney U test, *p* = 0.002, Figure 9D) and in advanced tumor stage (Mann–Whitney U test, I/II versus III/IV, *p* < 0.001, Figure 9B,C,E). On the other hand, both type A/AB thymomas and TCs displayed higher levels of nuclear TEAD4 expression compared to thymomas type B1, B2, and B3 (Kruskal–Wallis ANOVA, *p* = 0.005, Figure 9B–D). Nuclear TEAD4 expression was not correlated with patients’ overall survival (log-rank test, <15% versus ≥15%, *p* = 0.850, Table A10). However, cytoplasmic TEAD4 expression was correlated with worse patient overall survival (log-rank test, <70% versus ≥70%, *p* = 0.003, Figure 10, Table A11). Moreover, cytoplasmic TEAD4 expression was positively correlated with the patient’s age (Spearman correlation’s coefficient, R = 0.25, *p* = 0.027, Table A10). The associations of nuclear or cytoplasmic TEAD4 with the remaining clinicopathological parameters are presented in Table A10 and Table A11.

### 2.9. Associations Between the Investigated Molecules of the Hippo Cascade

Table 2 illustrates in detail the associations between the investigated molecules. There were positive associations between cytoplasmic MST1 with many of its downstream proteins, namely cytoplasmic LATS1, cytoplasmic MOB1A, cytoplasmic TAZ and cytoplasmic TEAD4 expression. The correlation with cytoplasmic SAV1 was only of marginal significance. Cytoplasmic SAV1 expression was positively correlated with cytoplasmic LATS1, whereas cytoplasmic LATS1 was also positively associated with cytoplasmic MOB1A and all the downstream proteins, namely nuclear and cytoplasmic TAZ, cytoplasmic YAP1, cytoplasmic AYAP, and nuclear and cytoplasmic TEAD4. Cytoplasmic MOB1A was also positively correlated with its downstream proteins cytoplasmic TAZ, cytoplasmic YAP1 and AYAP and cytoplasmic TEAD4. Both nuclear and cytoplasmic TAZ were positively associated with the downstream molecules, such as nuclear YAP1 as well as nuclear and cytoplasmic AYAP, whereas additionally nuclear TAZ showed positive correlations with nuclear and cytoplasmic TEAD4 with cytoplasmic TAZ being associated with cytoplasmic YAP1. As expected, nuclear and cytoplasmic YAP1 expression were positively associated with nuclear and cytoplasmic AYAP expression, and only cytoplasmic YAP1 expression had a significant positive correlation with nuclear and cytoplasmic TEAD4. On the other hand, only cytoplasmic AYAP was positively correlated with cytoplasmic TEAD4 expression.

## 3. Discussion

TETs, though extremely rare neoplasms, represent a significant clinical challenge due to their histological diversity, unpredictable behavior, and limited molecular characterization [7,10,13]. Despite the presence of specific staging systems like Masaoka-Koga, distinguishing indolent from aggressive TETs remains complex, as even histologically benign-appearing thymomas can exhibit invasive traits [10]. This underscores the need for molecular markers to refine prognostic stratification and guide therapeutic strategies. The Hippo signaling pathway, a conserved regulator of organ size, apoptosis, and cell proliferation, has emerged as a critical player in tumorigenesis and is thus a potential candidate in this regard [1,13,15,16,18]. However, its role in TETs remains poorly understood, with only one study by Palamaris et al. (2023) investigating some of the Hippo pathway components, namely YAP, TAZ, LATS1, and TEAD4 in TETs [1]. Building on this foundation, our study expands the scope by evaluating the additional upstream regulators MOB1A, MST1, and SAV1 alongside YAP, TAZ, LATS1, and TEAD4. Importantly, we also assess active (dephosphorylated) YAP, a functional marker reflecting Hippo pathway suppression and oncogenic potential [1,17].

Our analysis revealed distinct expression patterns of Hippo pathway components in TETs, possibly reflecting a complex and potentially context-dependent regulatory network. YAP1 showed predominantly a nuclear but also a cytoplasmic localization, with a positive correlation between the expression levels in both compartments. This predominantly nuclear localization aligns with the findings of Palamaris et al. (2023) and supports the assumption of Hippo pathway inactivation in TETs. Similar patterns have been reported in lung, breast, and gastrointestinal carcinomas [1,23]. Importantly, we are the first to report immunohistochemical detection of AYAP in TETs. AYAP was found both in the nucleus and in the cytoplasm in the majority of the examined cases, suggesting an ongoing activation of YAP transcriptional programs. This dual localization has also been described in breast cancer [24], but its biological relevance is not clear. TAZ showed nuclear expression in most of the cases, with half of them also displaying cytoplasmic immunoreactivity. These results are also consistent with Palamaris et al. (2023) and with findings in multiple tumor types such as non-small-cell lung cancer (NSCLC), hepatocellular, and colorectal carcinomas, as well as glioblastoma and breast cancer [1,25]. TEAD4, the main nuclear binding partner of YAP/TAZ, showed a predominantly cytoplasmic immunolocalization and an additional moderate nuclear expression. A similar cytoplasmic expression of TEAD4 was also reported by Palamaris et al. [1], which contrasts with TEAD4’s canonical role as a nuclear transcription factor [1,17]. However, cytoplasmic and even mitochondrial localization of TEAD4 has been reported in various cell types [26,27], suggesting broader, possibly context-dependent functions. Mechanistically, cytoplasmic accumulation of TEAD4 in TETs may result from upstream signaling events, such as activation of the MST1–Akt1–mTOR [28,29] or p38 stress pathways [30]. Alternatively, dominant-negative TEAD4 splice variants, which exhibit dual localization and lack transcriptional activity, might explain the cytoplasmic retention observed in tumors [31].

In normal thymic tissue, TEAD4 expression was predominantly nuclear, while cytoplasmic staining was only weakly detectable. This indicates that the pronounced cytoplasmic localization observed in TETs likely could represent a tumor-specific alteration rather than a physiological phenomenon. The specificity of the antibody used (Invitrogen PA5-21977, targeting amino acids 1 and 260 of TEAD4) is supported by prior validation in placental tissue, where a similar nuclear and cytoplasmic staining pattern was observed. In our study, lymphocytes remained consistently negative, and staining patterns were reproducible across different cases, further supporting the validity of the observed localization. Taken together, these findings suggest that cytoplasmic TEAD4 in TETs may not merely reflect passive mislocalization, but could point to alternative, non-transcriptional functions or a dysregulation of nuclear transport mechanisms. Whether this cytoplasmic presence represents a functional adaptation contributing to tumorigenesis, or a consequence of disrupted upstream regulation, remains to be determined. Further studies are needed to clarify whether TEAD4 engages in signaling roles outside the nucleus, potentially involving mitochondrial function, oxidative phosphorylation (OXPHOS) [30], or cytoplasmic signaling networks such as PI3K/AKT.

The upstream regulators of the Hippo pathway, namely MST1, SAV1, LATS1, and MOB1A, were almost exclusively localized to the cytoplasm, consistent with their canonical roles in Hippo signaling [17]. This pattern was particularly evident for MST1, which exhibited cytoplasmic staining in the vast majority of the cases, aligning with findings in pancreatic cancer cells [32]. Interestingly, one case showed nuclear MST1 expression, which may reflect apoptosis-related signaling, as such translocation has been linked to caspase-mediated cleavage in pancreatic and HER2-positive breast cancer [33,34,35,36]. In contrast to our observations, LATS1 was reported to be mainly nuclear in TETs in the study of Palamaris et al. (2023); however, cytoplasmic LATS1 expression is supported by studies in renal and colorectal carcinoma [1,37]. The cytoplasmic immunolocalization of SAV1 and MOB1A is also in line with previous reports in pancreatic carcinoma for SAV1 [17,38] and in NSCLC for MOB1A [39].

Positive staining in lymphocytes was observed for some upstream Hippo pathway components, namely LATS1 (35.1%), SAV1 (22.1%), and MOB1A (39.5%). While the primary focus of Hippo signaling lies in epithelial tumor cells, the presence of pathway components in tumor-infiltrating lymphocytes may suggest additional immunological roles. LATS1, SAV1, and MOB1A are known to regulate T-cell function, differentiation, and survival [40]. Their expression in the lymphocytic compartment could indicate local immune modulation, aberrant thymocyte selection, or activation states induced by the tumor microenvironment. This finding may be especially relevant in lymphocyte-rich thymoma subtypes and warrants further investigation to understand whether Hippo signaling in these immune cells contributes to immune surveillance, suppression, or escape mechanisms within the TET niche.

In our study, we found a moderated cytoplasmic MST1 staining in normal thymic tissue in about 50% of thymocytes, with occasional positivity in macrophages and lymphocytes. On the contrary, neoplastic tissue showed significantly stronger MST1 expression, suggesting an increased Hippo pathway activity in tumors. The same applied to LATS1, MOB1A, TAZ and TEAD4 and was less obvious but still apparent in the case of YAP1, all of which displayed an increased expression in neoplastic compared to normal tissue. On the other hand, AYAP expression was more prominent in normal compared to tumor tissue, while SAV1 was the only marker showing strong staining in both normal and neoplastic tissue. These findings highlight a potential dysregulation of Hippo signaling in thymomas, with variations in expression patterns that could correlate to tumorigenesis.

Nuclear YAP1 and AYAP expression was significantly higher in type A and AB thymomas compared to TCs, a result that is in line with Palamaris et al., who similarly found elevated nuclear YAP levels in these tumors [1]. Notably, both studies observed that B3 thymomas also exhibit increased nuclear YAP levels, underscoring the potential role of transcriptionally active YAP in these tumors. Accordingly, cytoplasmic MST1 expression as well as nuclear and cytoplasmic LATS1 expression were significantly higher in TCs and B3 thymomas. We also found that nuclear TAZ expression is highest in TETs with an A component and significantly lower in B1–B3 thymomas, while cytoplasmic TAZ is markedly diminished in B3 thymomas and TCs, suggesting that alterations in TAZ localization may correlate with tumor aggressiveness. In the same context, TEAD4 emerges as a critical marker since both nuclear and cytoplasmic TEAD4 are significantly high in TCs. These findings align with Palamaris et al. and implicate distinct roles of YAP1, AYAP, TAZ, MST1, LATS1, and TEAD4 in TET biology, suggesting that altered Hippo signaling may drive behavioral differences across histological types [1].

Our analysis revealed significant associations between the expression levels of several Hippo pathway components and the Masaoka–Koga tumor stage, further indicating a potential involvement in tumor aggressiveness. Cytoplasmic MST1, LATS1, MOB1A and TEAD4 expression was significantly elevated in advanced Masaoka–Koga stages compared to early stages. The association of LATS1 and TEAD4 with tumor stage has also been reported by Palamaris et al. [1]. These results point to associations between MST1, LATS1, MOB1A, and TEAD4 expression patterns and tumor stage, which may reflect changes in subcellular dynamics during progression. However, their exact functional contributions remain to be clarified. The observed cytoplasmic shift might be associated with reduced nuclear activity and a tendency toward YAP/TAZ activation in more aggressive tumors. In alignment with this hypothesis is the higher cytoplasmic YAP1 expression and reduced nuclear AYAP expression in advanced-stage TETs. Similarly, SAV1 showed a stage-dependent expression pattern: all advanced-stage tumors exhibited uniform positivity of SAV1 in all tumor cells, whereas in 23.4% of early-stage tumors, tumor cell subpopulations lacked SAV1 expression. This observation may point to a shift in Hippo pathway dynamics during tumor progression, yet the regulatory mechanisms underlying these patterns are still undefined and call for mechanistic clarification through future studies. Similar associations of the altered expression of members of the Hippo cascade with tumor stage have also been reported in other tumor types, such as NSCLC [18,41], prostate [41], pancreatic [42], colon and gastric [42,43], hepatocellular [15] and breast cancer [41,43].

One of the most compelling findings of our study is the identification of a significant association between high cytoplasmic TEAD4 expression and poorer overall survival in patients with TETs. While the majority of Hippo pathway components—including MST1, SAV1, LATS1, MOB1A, TAZ, YAP1, AYAP, and nuclear TEAD4—showed no prognostic impact, the cytoplasmic TEAD4 fraction emerged as a potential prognostic marker. As discussed above, cytoplasmic localization of TEAD4 may result from alternative splicing, disrupted nuclear transport, or upstream signaling alterations, and may reflect either a loss of transcriptional activity or the acquisition of non-nuclear oncogenic functions [25,26,27,28,29,30,31]. Although the underlying mechanisms remain speculative, our data suggest that the subcellular redistribution of TEAD4 may not be merely incidental possibly holding biological and clinical relevance.

Importantly, this is the first study to report cytoplasmic TEAD4 expression—independent of nuclear localization—as a prognostically relevant feature in TETs based on immunohistochemical analysis. While nuclear TEAD4 has previously been linked to prognosis in other malignancies, such as bladder cancer [44,45], renal clear cell carcinoma [46], and lung adenocarcinoma [47], and one glioma study applied a combined nuclear–cytoplasmic score [48], cytoplasmic TEAD4 has not previously been evaluated as an isolated prognostic variable. These findings may inform future research into non-canonical Hippo signaling and encourage exploration of TEAD4-targeting strategies, especially in tumors characterized by cytoplasmic accumulation. Nevertheless, validation in larger, well-characterized cohorts and functional studies will be necessary to confirm the prognostic and therapeutic relevance of this observation.

Correlation analysis revealed a coordinated expression pattern among Hippo pathway proteins in TETs, reflecting both canonical signaling architecture and tumor-specific regulatory adaptations. Cytoplasmic MST1 showed significant associations with downstream targets such as LATS1, MOB1A, TAZ, and TEAD4, indicating preserved upstream activity. Notably, LATS1 emerged as a central node, correlating positively with multiple downstream components—including TAZ, YAP1, AYAP, and TEAD4—supporting its key role in phosphorylating and retaining YAP/TAZ in the cytoplasm [16,49]. In contrast, SAV1 showed only weak associations, mainly with LATS1, and no clear links to other Hippo components. This limited connectivity deviates from its classical role as a scaffold protein within the MST–LATS complex [49]. Several factors may explain this finding, such as tumor-specific loss of regulatory function, post-translational modifications [50], or a more structural, non-dynamic role in TETs. Among downstream effectors, YAP1 and AYAP exhibited strong intra- and intercompartmental correlations, particularly in the nucleus. TEAD4 showed the strongest associations in the cytoplasm, correlating with cytoplasmic TAZ, LATS1, and AYAP. These findings suggest TEAD4 may engage in cytoplasmic interactions in TETs, possibly through complex formation, alternative splicing (e.g., TEAD4-S) [31], or cytoplasmic signaling crosstalk, unlike in other tumors such as hepatocellular [1,25] or colorectal carcinomas [19], where TEAD4 functions predominantly in the cell nucleus. Importantly, the lack of correlation between nuclear YAP1 and TEAD4 expression may indicate a divergence in their regulatory patterns, raising questions about their regulatory role, which should be addressed experimentally. Taken together, our results indicate that thymic tumors may employ alternative mechanisms of Hippo signaling modulation.

Several limitations of this study should be acknowledged. First, although the cohort size is reasonable for a rare tumor entity such as TETs, it remains limited and precludes multivariate survival analyses to assess the potential prognostic value of TEAD4 expression while adjusting for relevant clinical and pathological cofactors. Moreover, survival data were only available for a subset of patients, further limiting the statistical power and generalizability of outcome-related findings. Second, while immunohistochemical analysis provides valuable spatial and semi-quantitative insights into protein expression and localization, it is inherently limited by variability in antibody specificity, staining conditions, and scoring subjectivity [51]. Although internal controls and established evaluation criteria were applied, technical variation cannot be entirely excluded [52]. Third, the study was restricted to protein-level analyses and did not incorporate complementary molecular data such as mRNA expression or protein isoform analysis, which could provide a more detailed understanding of pathway activation, mechanistic relationships and functional relevance [53]. Fourth, potential differences in pre-analytical variables must be considered. Although all tissue samples underwent standardized fixation and storage procedures in an accredited histopathology laboratory, the retrospective study design spanning a period of ten years inherently carries the risk of subtle variations in tissue processing and long-term preservation. For all immunohistochemical analyses, antibody dilutions were selected based on manufacturer recommendations and standardized titration protocols, and staining was performed in a certified and accredited immunohistochemistry laboratory using a consistent detection system throughout. The only deviation from the standard protocol occurred with SAV1, where antigen retrieval conditions were optimized by extending the primary antibody incubation time to nine hours due to initially absent staining. This adjustment resulted in more intense signal, but internal negative tissue areas consistently served as controls. To reduce inter-observer variability, all stainings were scored by the same experienced pathologist. Antibodies that yielded no interpretable or absent staining results under validated conditions were categorically excluded from further analysis. Additionally, the study focused on a static analysis of protein expression in formalin-fixed, paraffin-embedded (FFPE) tissues, which do not capture dynamic signaling events or functional interactions. As such, the correlations observed cannot be interpreted as causal or mechanistically definitive. Finally, although the data suggest potential associations with histological subtypes and tumor stage, we refrained from making clinical claims or therapeutic implications due to the exploratory nature of the study and the lack of independent validation. Despite these limitations, the study provides novel and comprehensive insights into the expression and compartmental regulation of Hippo pathway components in TETs and lays a foundation for future mechanistic and translational research.

Building on these findings, future work should focus on dissecting the underlying biology and assessing clinical applicability. Functional validation of key regulators—particularly YAP, TAZ, and TEAD4—could be pursued in in vitro models derived from TET cell lines or engineered systems [41], in an effort to clarify possible causal links between subcellular localization, post-translational modifications, and transcriptional output. To complement immunohistochemical and functional in vitro data, quantitative gene expression analysis using Reverse Transcription quantitative Polymerase Chain Reaction (RT-qPCR) could serve as a valuable method to validate the observed protein-level differences—particularly for YAP, TAZ, and TEAD4—on the mRNA level, and to explore their correlation with tumor subtype or stage. In addition, integration of immunohistochemical data with RNA sequencing, phosphoproteomics, and spatial transcriptomics could provide a deeper understanding of active pathway states and reveal transcriptional programs driving tumor behavior [54,55]. Whole-exome or targeted Next-Generation Sequencing (NGS) might identify Hippo-related mutations or co-activated oncogenic pathways (e.g., PI3K/AKT, WNT) [56], potentially yielding novel combinatorial treatment strategies [57,58,59]. Integrating expression analysis with DNA-based NGS may clarify if genomic alterations drive Hippo dysregulation in TETs, shedding light on genotype–phenotype links. Importantly, the identification of cytoplasmic TEAD4 expression as a marker associated with poor prognosis highlights a potential therapeutic vulnerability. Recent developments in TEAD-targeting agents—including autopalmitoylation inhibitors (e.g., VT3989, IK-930) [60] may hold promise in YAP/TAZ-driven tumors. The distinct subcellular localization patterns of TEAD4 observed in our study raise important questions regarding the therapeutic modulation of the YAP/TEAD axis. While most current small-molecule inhibitors under clinical investigation aim to broadly disrupt the YAP–TEAD interaction (pan-TEAD inhibition) [22,61], our findings suggest that TEAD4 may exert context-dependent functions that differ between its nuclear and cytoplasmic fractions. In this light, isoform-specific or compartment-targeted strategies could potentially offer a more precise therapeutic approach. TEAD transcription factors are known to exist in multiple isoforms generated through alternative splicing [31], some of which may differ in their transcriptional activity, cofactor binding, or subcellular localization. These isoform-specific properties raise the possibility that broad-spectrum TEAD inhibition could affect both tumor-promoting and non-oncogenic isoforms, potentially limiting therapeutic specificity and increasing off-target effects. Isoform-selective inhibition, on the other hand, may allow for a more refined modulation of TEAD activity—especially in tumors where distinct TEAD isoforms exhibit divergent cellular functions. From a drug development perspective, the challenge lies in designing compounds that selectively inhibit TEAD isoforms or selectively target TEAD activity in particular cellular compartments, such as the cytoplasm or nucleus. Notably, recent studies have demonstrated that allosteric pan-TEAD inhibitors such as GNE-7883 not only suppress YAP/TAZ-mediated transcription but also overcome resistance to targeted therapies—including KRAS-G12C inhibitors—by blocking compensatory YAP/TAZ activation [62]. Additional compounds currently in clinical development have shown similar capabilities. VT104 (also known as VT3989), an allosteric TEAD inhibitor targeting the TEAD lipid pocket, demonstrated strong preclinical efficacy in models of KRAS-mutant cancers and is currently undergoing clinical evaluation [63]. Furthermore, IAG933, a first-in-class direct disruptor of the YAP–TEAD interface, has shown rapid and potent suppression of TEAD-dependent transcription and robust anti-tumor effects across Hippo-dysregulated and KRAS- or MAPK-altered tumor models—including mesothelioma, NSCLC, and pancreatic ductal adenocarcinoma—and is also in phase I clinical trials [64].

These findings emphasize the broader applicability of TEAD inhibition in both primary Hippo-dysregulated tumors and resistant tumor states. Whether patients with high cytoplasmic TEAD4 expression might benefit from such therapies remains an important question for future translational research.

In summary, these findings underscore the potential relevance of cytoplasmic TEAD4 in TET biology and prognosis and call for further mechanistic and translational investigations to explore its value as a biomarker and therapeutic target.

## 4. Materials and Methods

### 4.1. Patients

This is a study of archival FFPE tissue retrieved from the archives of the Department of Pathology, Nuremberg hospital from 77 patients with the diagnosis of a TET between 2013 and 2023 and available medical records. Table 3 presents patients’ characteristics in detail. 39 of the patients were men (50.6%) and 38 women (49.4%), with a median age at the time of diagnosis 69 years (range 21–88 years). The relative frequency of WHO subtypes was as follows: type A 3.9%; type AB 33.7%; type B1 9%; type B2 14.3%; type B3 18.2%; micronodular thymoma with lymphoid stroma (MNT) 2.6%; and thymic carcinoma (TC) 18.2%. Masaoka–Koga stage was I in 33.8%; II in 38.5% III in 13.8%; IVa in 7.7%; IVb in 6.1% of the patients. Surgical margins were positive in 18.7% of the cases, for which this information was available. Co-existing myasthenia gravis was diagnosed in 11 patients. Follow-up information was available for 60 patients, ranging from 0.3 to 109.4 months (median: 21.8 months).

### 4.2. Immunohistochemistry

Immunohistochemistry (IHC) for the molecules of the Hippo cascade was performed on formalin-fixed, paraffin-embedded tissue sections. From each paraffin block, a 2 µm-thick section was obtained, mounted on a slide, and allowed to dry at 37 °C. Subsequently, the sections were stained with hematoxylin and eosin using an automated staining system (Autostainer Link48, Dako [Agilent Technologies, Santa Clara, CA, USA] in combination with the EnVision FLEX Kit, Agilent Technologies, Santa Clara, CA, USA). The following antibodies were used for the IHC: YAP1 (mouse monoclonal (1A12), Invitrogen [Thermo Fisher Scientific, Waltham, MA, USA], 1:1000); Anti-active YAP1 (rabbit monoclonal (EPR19812), Abcam, Cambridge, UK, 1:2000); WWTR1 Antibody (TAZ) (mouse monoclonal (2A12A10), Proteintech, Planegg-Martinsried, Germany, 1:600); LATS1 (rabbit polyclonal, Proteintech, Planegg-Martinsried, 1:200); MOB1A (rabbit polyclonal, Invitrogen [Thermo Fisher Scientific, Waltham, MA, USA], 1:350); SAV1 (mouse monoclonal (OTI2B7), Invitrogen [Thermo Fisher Scientific, Waltham, MA, USA], 1:100); MST1 (rabbit polyclonal, Invitrogen [Thermo Fisher Scientific, Waltham, MA, USA], 1:250); TEAD4 (rabbit polyclonal, Invitrogen [Thermo Fisher Scientific, Waltham, MA, USA], 1:500). For positive controls, we selected tissue samples known to express the target proteins. Specifically, prostate tissue was used for YAP1, human breast cancer tissue for anti-active YAP1, TAZ, LATS1, and MOB1A, placenta tissue for TEAD4, human kidney tissue for SAV1, and human adenocarcinoma tissue for MST1. These positive controls were included in every staining run. For each antibody, normal tissue was co-stained or evaluated adjacent to tumor tissue on the same slide, enabling a direct comparison between normal and neoplastic regions.

Antibody validation was performed by titrating each antibody within the dilution range recommended by the manufacturer, as suggested by the respective datasheets. The selected dilutions provided strong, specific signals with minimal background. While no separate negative control tissues were processed in parallel, we consistently assessed areas within each tissue section that are expected to be negative for the respective marker, such as stromal or vascular regions, particularly in cases with intense overall staining (e.g., SAV1). This approach helped confirm the specificity of the observed signals and avoid false-positive interpretation due to non-specific binding. Observed subcellular localization patterns (e.g., nuclear, cytoplasmic) were critically assessed and compared to literature and manufacturer references to ensure staining plausibility. The antibody datasheets (Appendix A) and detailed validation protocols (Appendix A), including titration series and evaluation criteria, are provided in the Appendix A.

IHC evaluation was performed by counting at least 1000 tumor cells in each case independently by an experienced pathologist (L.G.), blinded to clinical information. Nuclear and cytoplasmic immunoreactivity was evaluated separately. The extent of protein expression was calculated by the percentage of positive tumor cells to the total number of tumor cells within each specimen. The staining intensity was estimated in four categories: 0 (no reaction), 1 (mild reaction), 2 (moderate reaction), and 3 (intense reaction).

### 4.3. Statistical Analysis

Statistical analysis was performed by a MSc biostatistician (L.G.). The association between the IHC expression of MST1, MOB1A, SAV1, YAP1, AYAP, TAZ, LATS1, and TEAD4 with clinicopathological characteristics was examined using non-parametric tests with correction for multiple comparisons, as appropriate. Survival analysis was performed using Kaplan–Meier survival curves and the differences between the curves were compared with log-rank test. Numerical parameters were categorized based on the median value. ROC (Receiver Operating Characteristic) analysis confirmed these thresholds. Since TETs belong to the group of rare diseases, traditional sample size analysis was not performed. To secure the statistical validity of our results, however, we conducted power analysis of the performed statistical tests, which showed that in each case a power of more than 0,85 was reached (SPSS Statistics for Windows, version 21.0, SPSS Inc., Chicago, IL, USA). Due to the small number of cases and events in our cohort, multivariate statistical analysis, including multivariate survival analysis, was not conducted. A *p*-value of <0.05 was considered statistically significant. The analysis was performed with the statistical package STATA 11.0/SE (College Station, TX, USA) for Windows.

## 5. Conclusions

Our findings demonstrate that the core components of the Hippo signaling pathway are differentially expressed and compartmentalized in TETs, being also differentially expressed in neoplastic compared to normal tissue and showing significant variations according to histological subtype and tumor stage. In particular, these findings indicate a retained or possibly reactivated Hippo pathway activity in advanced tumors, potentially reflecting a shift toward non-canonical, TEAD-independent mechanisms of tumor progression. Importantly, cytoplasmic TEAD4 expression emerged as a novel prognostic marker associated with reduced overall survival, highlighting its potential role in alternative signaling pathways beyond the classical Hippo axis. Future studies should incorporate functional assays and integrative approaches such as single-cell omics, spatial transcriptomics, and targeted expression analyses by RT-qPCR, as well as DNA-based sequencing strategies, to further elucidate the biological, genomic, and immunological roles of Hippo signaling in thymic tumorigenesis.

## Figures and Tables

**Figure 1 ijms-26-05938-f001:**
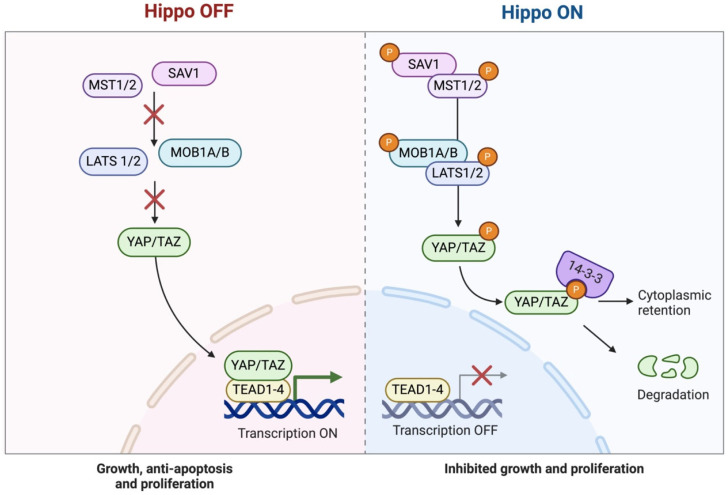
Hippo signaling pathway. Created in BioRender. Elm, L. (2025); https://BioRender.com/3ebfg0o (accessed on 15 May 2025).

**Figure 2 ijms-26-05938-f002:**
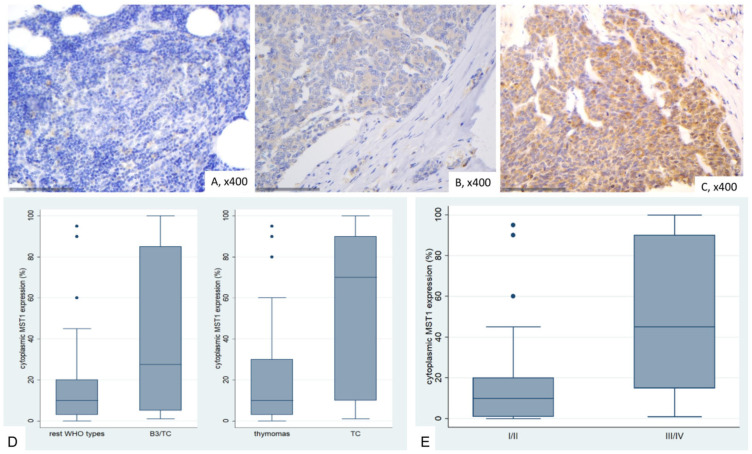
Immunohistochemical expression of MST1 in TETs. (**A**) Normal thymic tissue (cytoplasmic; ×400); (**B**) Type A thymoma (cytoplasmic; ×400); (**C**) Thymic carcinoma (cytoplasmic; ×400); (**D**) Associations of MST1 expression with WHO histological type; (**E**) associations of MST1 expression with Masaoka-Koga stage.

**Figure 3 ijms-26-05938-f003:**
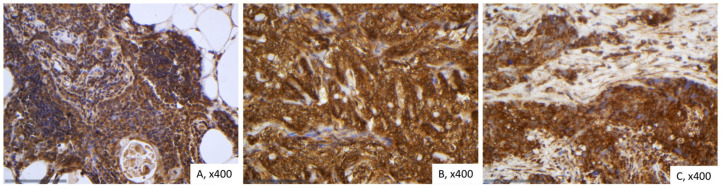
Immunohistochemical expression of SAV1 in TETs. (**A**) Normal thymic tissue (cytoplasmic; ×400); (**B**) Type AB thymoma with negative endothelial cells serving as negative internal control (cytoplasmic; ×400); (**C**) Thymic carcinoma (cytoplasmic; ×400).

**Figure 4 ijms-26-05938-f004:**
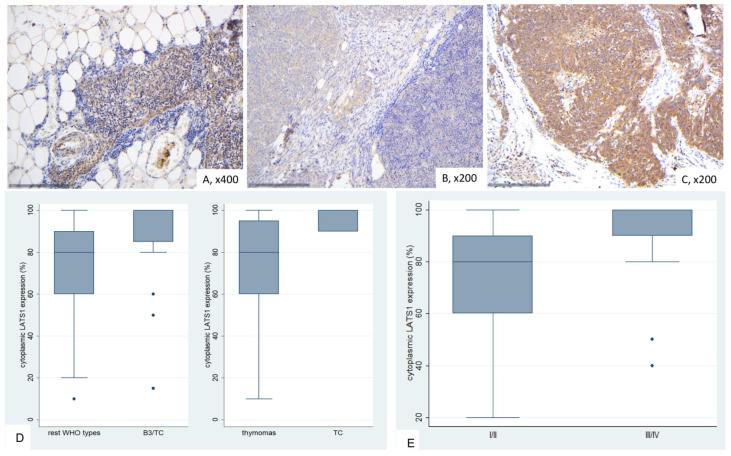
Immunohistochemical expression of LATS1 in TETs. (**A**) Normal thymic tissue (cytoplasmic, nuclear; ×400); (**B**) Type AB thymoma (cytoplasmic; ×200); (**C**) Thymic carcinoma (cytoplasmic; ×200); (**D**) associations of LATS1 expression with WHO histological type; (**E**) Associations of LATS1 expression with Masaoka-Koga stage.

**Figure 5 ijms-26-05938-f005:**
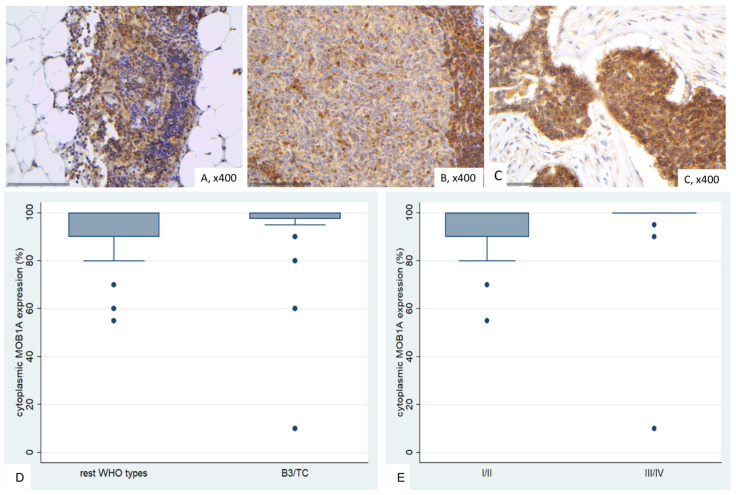
Immunohistochemical expression of MOB1A in TETs. (**A**) Normal thymic tissue (cytoplasmic; ×400); (**B**) Type A thymoma (cytoplasmic; ×400); (**C**) Thymic carcinoma (cytoplasmic; ×400); (**D**) associations of MOB1A expression with WHO histological type; (**E**) associations of MOB1A expression with Masaoka-Koga stage.

**Figure 6 ijms-26-05938-f006:**
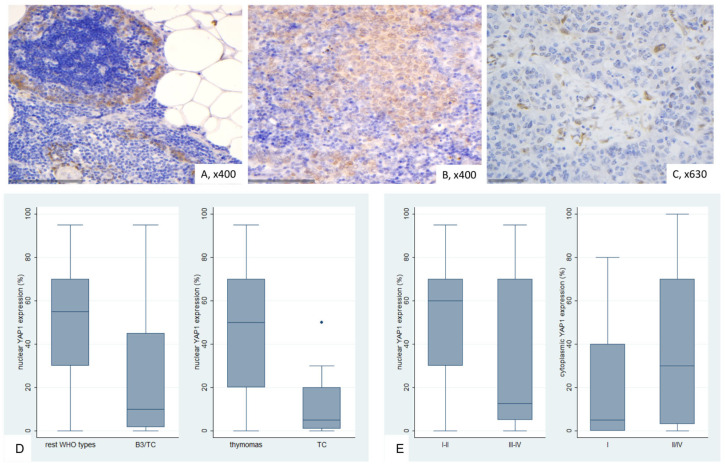
Immunohistochemical expression of YAP1 in TETs. (**A**) Normal thymic tissue (cytoplasmic, nuclear; ×400); (**B**) Type B1 thymoma (cytoplasmic; ×400); (**C**) Thymic carcinoma (nuclear; ×630); (**D**) associations of nuclear YAP1 with WHO histological type; (**E**) associations of nuclear and cytoplasmic YAP1 with Masaoka-Koga stage.

**Figure 7 ijms-26-05938-f007:**
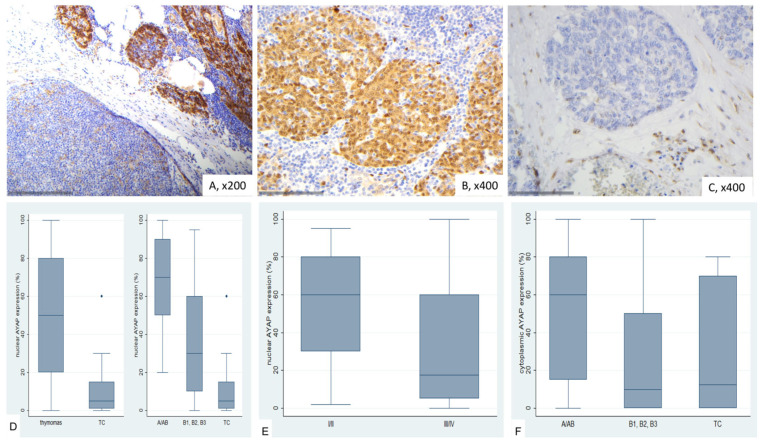
Immunohistochemical expression of active YAP1 (AYAP) in TETs. (**A**) Type B2 thymoma next to normal thymic tissue (nuclear, cytoplasmic; ×200); (**B**) Type A thymoma (nuclear, cytoplasmic; ×400); (**C**) Thymic carcinoma (nuclear; ×400); (**D**) associations of nuclear AYAP expression with WHO histological type; (**E**) associations of nuclear AYAP expression with Masaoka-Koga stage; (**F**) associations of cytoplasmic AYAP expression with WHO histological type.

**Figure 8 ijms-26-05938-f008:**
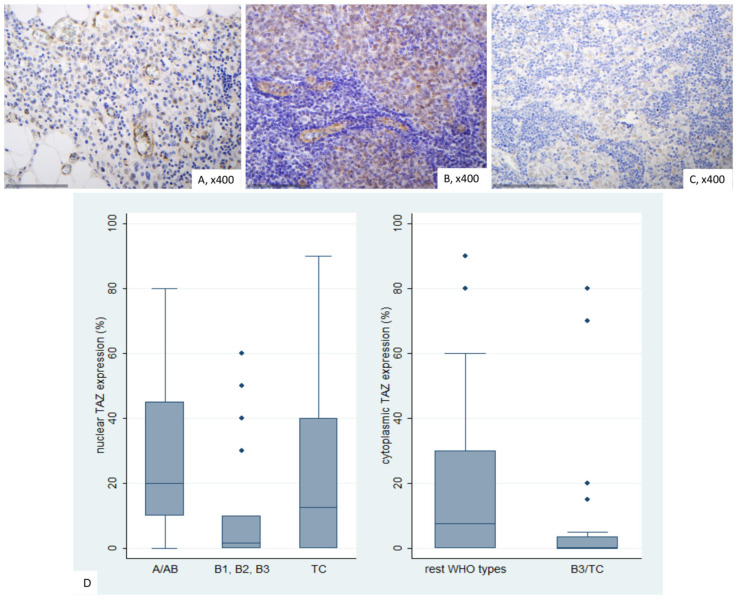
Immunohistochemical expression of TAZ in TETs. (**A**) Normal thymic tissue (cytoplasmic, nuclear; ×400); (**B**) Type A thymoma (cytoplasmic, nuclear; ×400); (**C**) Type B2 thymoma (nuclear; ×400); (**D**) associations of nuclear and cytoplasmic TAZ expression with WHO histological type.

**Figure 9 ijms-26-05938-f009:**
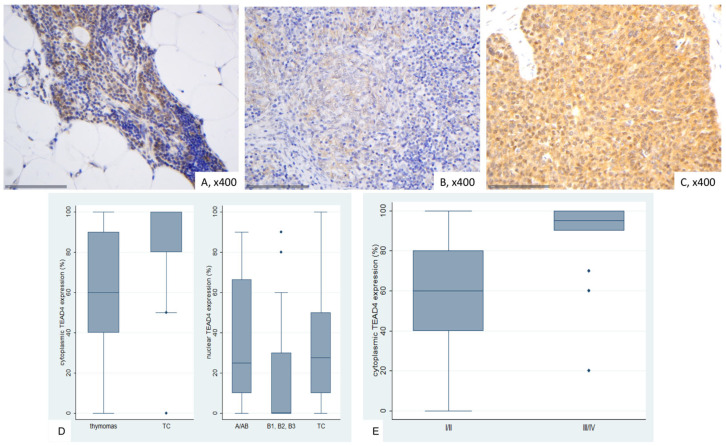
Immunohistochemical expression of TEAD4 in TETs. (**A**) Normal thymic tissue (nuclear, cytoplasmic; ×400); (**B**) Type AB thymoma (cytoplasmic; ×400); (**C**) Thymic carcinoma (cytoplasmic, nuclear; ×400); (**D**) association of nuclear and cytoplasmic TEAD4 expression with WHO histological type; (**E**) association of cytoplasmic TEAD4 expression with Masaoka-Koga stage.

**Figure 10 ijms-26-05938-f010:**
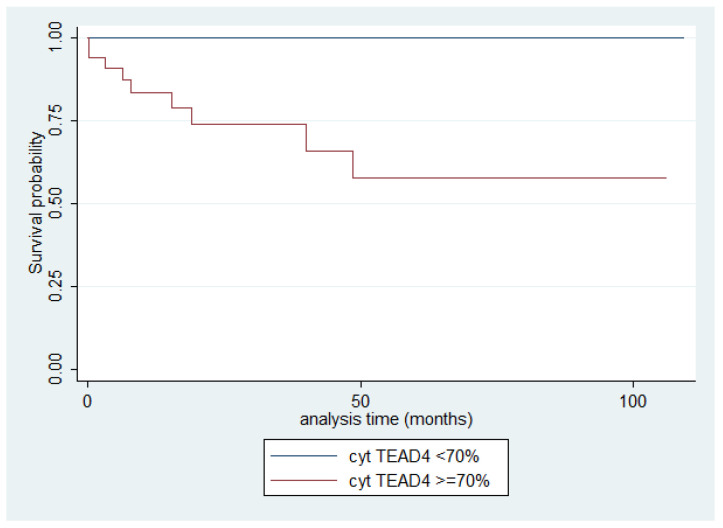
Kaplan–Meier survival curves for the cytoplasmic TEAD4 expression in TETs.

**Table 1 ijms-26-05938-t001:** Summary of immunohistochemical findings for all investigated Hippo pathway molecules in TETs, including subcellular localization, associations with tumor size, WHO histological classification, Masaoka-Koga stage, and survival.

Molecule	Immunolocalization	Tumor Size	WHO Histological Types	Masaoka-Koga Stage	Survival Analysis
MST1	Cytoplasmic (96.1%)	No significant correlation (Spearman’s correlation coefficient, *p* = 0.152)	Higher expression in B3/TC (Mann–Whitney U test, *p* = 0.032)	Higher expression in III/IV (Mann–Whitney U test, *p* = 0.002)	No significant correlation (log-rank, >10% versus ≥10%, *p* = 0.99)
SAV1	Cytoplasmic (100%)	No significant correlation (Spearman’s correlation coefficient, *p* = 0.237)	More uniformly high expression in B3/TC (Fisher’s exact test, *p* = 0.025)	Higher expression in III/IV (Fisher’s exact test, I/II versus III/IV, *p* = 0.027)	No significant correlation (log-rank, <100% versus 100%, *p* = 0.479)
LATS1	Cytoplasmic (100%)	No significant correlation (Mann–Whitney U test, *p* = 0.566)	Higher expression in B3/TC (Mann–Whitney U test, *p* < 0.001)	Higher expression in III/IV (Mann–Whitney U test, *p* = 0.007)	No significant correlation (log-rank, *p* = 0.838)
MOB1A	Cytoplasmic (100%)	No significant correlation (Mann–Whitney U test, *p* = 0.389)	Higher in B3/TC (Mann–Whitney U test, marginal, *p* = 0.063)	Higher in III/IV (Mann–Whitney U test, I/II versus III/IV, *p* = 0.033)	No significant correlation (log-rank, >100% versus 100%, *p* = 0.592)
YAP1	Nuclear (90.9%), Cytoplasmic (63.4%)	No significant correlation (Mann–Whitney U test, nuclear: *p* = 0.870; cytoplasmic: *p* = 0.144)	Lower nuclear expression in B3/TC (Mann–Whitney U test, *p* = 0.010)	Lower nuclear in III/IV (Mann–Whitney U test, I/II versus III/IV, *p* = 0.023); higher cytoplasmic in III/IV (Mann-Whitney U test, I versus II/III/IV, *p* = 0.032)	No significant correlation (nuclear: log-rank, <40% versus ≥40%, *p* = 0.323; cytoplasmic: <15% versus ≥15%, *p* = 0.468)
AYAP	Nuclear (93.5%), Cytoplasmic (75.3%)	Nuclear and cytoplasmic: no significant correlation (Mann–Whitney U test, *p* = 0.131)	Lower nuclear expression in TC (Mann–Whitney U test, *p* < 0.001); highest cytoplasmic expression in A/AB (Kruskal–Wallis ANOVA, *p* = 0.011)	Lower nuclear in III/IV (Mann–Whitney U test, I/II versus III/IV, *p* = 0.007)	No significant correlation (nuclear: log-rank, <40% versus ≥40%, *p* = 0.296; cytoplasmic: <30% versus ≥30%, *p* = 0.614)
TAZ	Nuclear (68.8%), Cytoplasmic (51.9%)	No significant correlation (Mann–Whitney U test, nuclear: *p* > 0.999; cytoplasmic: *p* = 0.123)	Nuclear higher in A/AB (Kruskal–Wallis ANOVA, *p* = 0.004); cytoplasmic lower in B3/TC (Mann–Whitney U test, *p* = 0.004)	No significant correlation (nuclear: Mann–Whitney U test, I/II versus III/IV, *p* = 0.182; cytoplasmic: Mann–Whitney U test, I/II versus III/IV, *p* = 0.281)	No significant correlation ( nuclear: log-rank, <10% versus ≥10%, *p* = 0.951; cytoplasmic: log-rank, <2% versus ≥2%, *p* = 0.218)
TEAD4	Nuclear (63.6%), Cytoplasmic (94.8%)	No significant correlation (Mann–Whitney U test, nuclear: *p* = 0.348; cytoplasmic: *p* = 0.222)	Higher cytoplasmic in TCs (Mann–Whitney U test, *p* = 0.002); higher nuclear in A/AB and TCs (Kruskal–Wallis ANOVA, *p* = 0.005)	Higher cytoplasmic in III/IV (Mann–Whitney U test, I/II versus III/IV, *p* < 0.001)	Nuclear: no siginifcant correlation (log-rank, <15% versus ≥15%, *p* = 0.850) Worse OS if cytoplasmic ≥70% (log-rank, *p* = 0.003)

**Table 2 ijms-26-05938-t002:** Associations between the investigated molecules of the Hippo cascade. Results of Spearman correlation’s coefficient. Bold decorates statistically significant associations. Positive correlations are shown in orange. The color intensity indicates the strength of the correlation.

	n- MST1	c-MST1	c-SAV1	c-LATS1	c-MOB1A	n-TAZ	c-TAZ	n-YAP1	c-YAP1	n-AYAP	c-AYAP	n-TEAD4
c-MST1	R = −0.19 *p* = 0.095											
c-SAV1	R = 0.048 *p* = 0.68	R = 0.19 *p* = 0.096										
c-LATS1	R = −0.14 *p* = 0.206	**R = 0.45** ***p* < 0.001**	**R = 0.33** ***p* = 0.003**									
c-MOB1A	R = 0.09 *p* = 0.441	**R = 0.24** ***p* = 0.032**	R = 0.05 *p* = 0.624	**R = 0.34** ***p* = 0.002**								
n-TAZ	R = 0.12 *p* = 0.275	R = −0.08 *p* = 0.459	R = −0.10 *p* = 0.366	**R = 0.25** ***p* = 0.025**	R = 0.02 *p* = 0.814							
c-TAZ	R = 0.13 *p* = 0.246	**R = 0.24** ***p* = 0.038**	R = 0.01 *p* = 0.919	**R = 0.22** ***p* = 0.056**	**R = 0.23** ***p* = 0.042**	R = 0.19 *p* = 0.097						
n-YAP1	R = −0.02 *p* = 0.816	R = −0.21 *p* = 0.070	R = −0.21 *p* = 0.064	R = −0.05 *p* = 0.631	R = 0.04 *p* = 0.729	**R = 0.30** ***p* = 0.007**	**R = 0.32** ***p* = 0.004**					
c-YAP1	R = −0.13 *p* = 0.244	R = 0.11 *p* = 0.340	R = −0.02 *p* = 0.857	**R = 0.35** ***p* = 0.001**	**R = 0.23** ***p* = 0.044**	R = 0.13 *p* = 0.239	**R = 0.27** ***p* = 0.016**	**R = 0.36** ***p* = 0.001**				
n-AYAP	R = 0.16 *p* = 0.16	R = −0.20 *p* = 0.07	R = −0.20 *p* = 0.080	R = −0.02 *p* = 0.826	R = 0.05 *p* = 0.653	**R = 0.37** ***p* = 0.001**	**R = 0.44** ***p* < 0.001**	**R = 0.80** ***p* < 0.001**	**R = 0.30** ***p* = 0.009**			
C-AYAP	R = 0.08 *p* = 0.497	R = −0.01 *p* = 0.872	R = −0.03 *p* = 0.788	**R = 0.29** ***p* = 0.010**	**R = 0.29** ***p* = 0.009**	**R = 0.28** ***p* = 0.013**	**R = 0.23** ***p* = 0.047**	**R = 0.45** ***p* < 0.001**	**R = 0.68** ***p* < 0.001**	**R = 0.58** ***p* < 0.001**		
n-TEAD4	R = −0.01 *p* = 0.867	R = 0.13 *p* = 0.265	R = −0.01 *p* = 0.892	**R = 0.36** ***p* = 0.001**	R = 0.01 *p* = 0.940	**R = 0.34** ***p* = 0.002**	R = 0.06 *p* = 0.600	R = 0.07 *p* = 0.526	**R = 0.24** ***p* = 0.038**	R = 0.07 *p* = 0.519	R = 0.13 *p* = 0.245	
c-TEAD4	R = 0.08 *p* = 0.468	**R = 0.31** ***p* = 0.005**	R = 0.13 *p* = 0.265	**R = 0.62** ***p* < 0.001**	**R = 0.29** ***p* = 0.012**	**R = 0.25** ***p* = 0.029**	R = 0.11 *p* = 0.318	R = −0.04 *p* = 0.706	**R = 0.27** ***p* = 0.017**	R = −0.07 *p* = 0.523	**R = 0.24** ***p* = 0.031**	**R = 0.31** ***p* = 0.007**

**Table 3 ijms-26-05938-t003:** Characteristics of 77 patients with TETs included in our analysis.

Parameter	Median	Min–Max
**Age (years)**	69	21–88
**Tumor size (cm)**	6.5	0.9–14
	**Number**	**%**
**Gender**		
Male	39	50.6
Female	38	49.4
**WHO subtypes**		
Type A	3	3.9
Type AB	26	33.7
Type B1	7	9
Type B2	11	14.3
Type B3	14	18.2
Micronodular with lymphoid stroma (MNT)	2	2.6
Thymic carcinoma (TC)	14	18.2
**Masaoka–Koga stage**		
I	22	33.8
II	25	38.5
III	9	13.8
IVa	5	7.7
IVb	4	6.1
**Presence of myasthenia Gravis**	11	14.1
**Event**		
Cencored-Alive	51/60, follow-up 0.3–109, 4 months	85
Dead	9/60, within 0.1–48.6 months	15

## Data Availability

The data in this work can be obtained upon request from the corresponding author.

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
