# Peer review of "Hippo Pathway Dysregulation in Thymic Epithelial Tumors (TETs): Associations with Clinicopathological Features and Patients’ Prognosis"

_ijms, 2025, doi:10.3390/ijms26135938_

Round 1
Reviewer 1 Report
Comments and Suggestions for Authors
This paper offers new and important information about the Hippo pathway in TETs, focusing on the role of cytoplasmic TEAD4 in predicting outcomes. The study is useful, but some parts need small changes to make it clearer and stronger.
- In the Abstract, it is important to clearly show what makes this study different from others. Also, adding at least one statistical detail is key to making the findings more accurate and believable.
- It might be beneficial to consider simplifying the longer and repetitive sentences, particularly around line 93.
It is very important to explain why the sample size is chosen, especially because TETs are rare. This helps make sure the study is valid and reliable.
The way different observers agree on results must be explained clearly. Saying "complete compliance" is not detailed enough and does not show the high standards needed for these analyses.
- Not correcting for multiple comparisons is a big mistake that needs fixing to keep the statistical analysis trustworthy. - Detailed information about how antibodies are checked and improved is needed to support the study's results and strengthen its scientific credibility. - Explaining why certain survival cutoff points are used is important, and discussing their limits will make the study more transparent and strong.
- The results section is full of details and data but needs some improvements. First, organize the information by showing location, size, strength, and importance. This will make it clearer. Next, add scale bars and magnification details to the images for accuracy. Explain the survival analysis cutoffs in each paragraph for better understanding. Summarize the findings in a table to avoid repeating information and make it easier to read. These changes will make the results section more effective.
- The discussion is detailed but sometimes exaggerates conclusions about mechanisms. To be more credible, it is important to present IHC data as ideas to explore, not final answers. This matches scientific standards and encourages more research. Be careful when talking about interactions with other pathways like mTOR and Akt to avoid making broad claims. The unclear parts about cytoplasmic TEAD4 need to be explained better to make the study stronger.
The limitations section should be expanded to talk about issues like not correcting for multiple comparisons, differences in pre-analysis, and limited survival data. This will help people understand the study's limits. Also, avoid making clinical claims without more proof to keep conclusions based on strong evidence. These steps will keep the study honest and make it more influential in the field.
- Add scale bars and magnification labels to IHC figures to provide context and enhance credibility. Improve correlation matrix readability through color coding.
Author Response
Comment 1: In the Abstract, it is important to clearly show what makes this study different from others. Also, adding at least one statistical detail is key to making the findings more accurate and believable.
Response 1: Thank you for pointing this out. We tried to modify the abstract according to the reviewer’s suggestions. Due to word amount limitation (200 words), however, we could not include extensive information regarding the statistical analysis except for the p-values and the cut-off used for TEAD4 analysis: "Cytoplasmic TEAD4 overexpression was associated with poorer overall survival (log-rank, <70% versus ≥70%, p = 0.003)." ("Abstract, page 1, lines 22-24). An explanation regarding the difference of this study compared to the previous ones is also provided ''We performed the first comprehensive immunohistochemical analysis of core and upstream Hippo pathway components—YAP1, active YAP (AYAP), TAZ, LATS1, MOB1A, MST1, SAV1, and TEAD4—in 77 TETs.'' ("Abstract", page 1, lines 13-15).
Comment 2: It might be beneficial to consider simplifying the longer and repetitive sentences, particularly around line 93.
Response 2: Thank you for your comment. We agree. We shortened some longer sentences, (particularly around line 93, as suggested by the reviewer). Our modifications are the following:
- "Tumor invasiveness is a more precise prognostic factor, as 15-year survival drops from 47% in non-invasive to 12.5% in invasive TETs" ("1. Introduction", page 2, lines 46-48)
- ''Distinguishing indolent from aggressive TETs remains difficult, highlighting the value of molecular markers for diagnosis and prognosis" ("1. Introduction", page 2, lines 49-51)
- "Although advances have been made in understanding TET biology, the Hippo pathway’s role and mechanisms of dysregulation remain poorly understood and insufficiently studied" ("1. Introduction."("1. Introduction", page 3, lines 85-87)
- "This study extends previous analyses by investigating both core components (YAP1, TAZ, LATS1, and TEAD4) and upstream regulators (MOB1A, MST1, and SAV1) of the Hippo pathway across different TET subtypes and their clinical associations. Additionally, we analyze expression correlations between these components to identify possible functional interactions and gain deeper insight into Hippo pathway regulation in TETs." ("1. Introduction", page 3, lines 91-95).
- "By examining these relationships, we aim to evaluate molecular markers that could potentially improve risk stratification, support personalized therapy, and enhance outcomes in these rare, challenging tumors" ("1. Introduction", page 3, lines 99-102)
- "The Hippo signaling pathway, a conserved regulator of organ size, apoptosis, and cell proliferation, has emerged as a critical player in tumorigenesis and is thus a potential candidate in this regard" ("3. Discussion, page 14, lines 349-352)
- "Interestingly, one case showed nuclear MST1 expression, which may reflect apoptosis-related signaling, as such translocation has been linked to caspase-mediated cleavage in pancreatic and HER2-positive breast cancer" ("3. Discussion, page 15/16, lines 404-406)
- "These findings align with Palamaris et al. and implicate distinct roles of YAP1, AYAP, TAZ, MST1, LATS1, and TEAD4 in TET biology, suggesting that altered Hippo signaling may drive behavioral differences across histological types" ("3. Discussion, page 16, lines 444-447).
- "Importantly, this is the first study to report cytoplasmic TEAD4 expression—independent of nuclear localization—as a prognostically relevant feature in TETs based on immunohistochemical analysis." ("3. Discussion", page 17, lines 480-483).
- "Integrating expression analysis with DNA-based NGS may clarify if genomic alterations drive Hippo dysregulation in TETs, shedding light on genotype–phenotype links" ("3. Disussion, page 19, lines 566-567).
Comment 3: It is very important to explain why the sample size is chosen, especially because TETs are rare. This helps make sure the study is valid and reliable.
Response 3: Thank you for pointing this out. We agree with that comment. In order to secure the validity of our results we conducted a power analysis of the performed statistical tests and we added the following sentence in the statistical analysis section: "Since TETs belong to the group of rare diseases, traditional sample size analysis was not performed. In order to secure the statistical validity of our results, however, we conducted power analysis of the performed statistical tests, which showed that in each case a power of more than 0,85 was reached (SPSS Statistics for Windows, version 21.0, SPSS Inc., Chicago, Ill., USA). Due to the small number of cases and events in our cohort a multivariate statistical analysis, including multivariate survival analysis was not conducted." ("4.3 Statistical analysis", page 21, lines 659-664).
Comment 4: The way different observers agree on results must be explained clearly. Saying "complete compliance" is not detailed enough and does not show the high standards needed for these analyses.
Response 4: Thank you for your valuable comment. Indeed, the term “complete compliance” is not appropriated. We corrected the respective sentence of the manuscript omitting this term.
Comment 5: Not correcting for multiple comparisons is a big mistake that needs fixing to keep the statistical analysis trustworthy. - Detailed information about how antibodies are checked and improved is needed to support the study's results and strengthen its scientific credibility. - Explaining why certain survival cutoff points are used is important, and discussing their limits will make the study more transparent and strong.
Response 5: Thank you for your comment. A correction of multiple comparisons was indeed made for all statistical tests in our analysis, a fact which due to a typographical error was not made clear in the original version of our manuscript. In particular, in the original version of our manuscript there was the following sentence “The association between the IHC expression of MST1 with clinicopathological characteristics was examined using non-parametric tests with correction for multiple comparisons, as appropriate”, although the same type of statistical analysis was performed for MOB1A, SAV1, YAP1, AYAP, TAZ, LATS1, and TEAD4. We corrected this sentence in the revised version of our manuscript accordingly “The association between the IHC expression of MST1, MOB1A, SAV1, YAP1, AYAP, TAZ, LATS1, and TEAD4 with clinicopathological characteristics was examined using non-parametric tests with correction for multiple comparisons, as appropriate." (“4.3 Statistical analysis”, page 21, lines 652-655).
As already mentioned in the same paragraph we used in all the investigated molecules the median value of each parameter as a cut-off for our statistical analysis (“4.3 Statistical analysis”, page 21, lines 657-658). We also now performed ROC (Receiver Operating Characteristic) analysis to confirm this cut-off in every case. The same cut-off for each molecule was used throughout the statistical analysis: "ROC (Receiver Operating Characteristic) analysis confirmed these thresholds" (“4.3 Statistical analysis”, page 21, lines 658-660).
Detailed information regarding the checking and testing the antibodies is now provided in the paragraph “4.2 Immunohistochemistry” ("4. Materials and Methods", page 17, lines 577-591): "These positive controls were included in every staining run. For each antibody, normal tissue was co-stained or evaluated adjacent to tumor tissue on the same slide, enabling a direct comparison between normal and neoplastic regions. Antibody validation was performed by titrating each antibody within the dilution range recommended by the manufacturer, as suggested by the respective datasheets. The selected dilutions provided strong, specific signals with minimal background. While no separate negative control tissues were processed in parallel, we consistently assessed areas within each tissue section that are expected to be negative for the respective marker, such as stromal or vascular regions, particularly in cases with intense overall staining (e.g. SAV1). This approach helped confirm the specificity of the observed signals and avoid false-positive interpretation due to non-specific binding. Observed subcellular localization patterns (e.g., nuclear, cytoplasmic) were critically assessed and compared to literature and manufacturer references to ensure staining plausibility. The antibody datasheets (S1) and detailed validation protocols (S2), including titration series and evaluation criteria, are provided in the Supplementary Materials." (“4.3 Statistical analysis”, page 21, lines 629-643).
Finally, the limitation regarding the small sample size is also discussed in the same paragraph ("4.3 Statistical analysis", page 21, lines 659-664) and in the "3. Discussion" section (page 18, lines 514-519): "First, although the cohort size is reasonable for a rare tumor entity such as TETs, it remains limited and precluded multivariate survival analyses to assess the potential prognostic value of TEAD4 expression while adjusting for relevant clinical and pathological cofactors. Moreover, survival data were only available for a subset of patients, further limiting the statistical power and generalizability of outcome-related findings."
Comment 6: The results section is full of details and data but needs some improvements. First, organize the information by showing location, size, strength, and importance. This will make it clearer. Next, add scale bars and magnification details to the images for accuracy. Explain the survival analysis cutoffs in each paragraph for better understanding. Summarize the findings in a table to avoid repeating information and make it easier to read. These changes will make the results section more effective.
Response 6: Thank you for your comment. However, in our opinion there is already a concept of organization the provided information: The antibodies in the results section are listed according to their position within the signaling pathway, from upstream to downstream and ends with the most clinically relevant transcription factor, TEAD4. This information is now provided in the article to make it easier for the reader to follow our results: "Our results are presented in a sequence according to the position of the investigated molecules within the signaling pathway, from upstream to downstream." ("2. Results", page 3, lines 107-109). The cut-offs used in the survival analysis are already explained in response 5. To give a better overview over our results, we added a comprehensive table added in the article: "To provide a concise overview of the major findings, Table 1 summarizes the expression patterns and localization of all investigated Hippo pathway components across the analyzed TET subtypes and stages. This summary serves as a guide to the more detailed descriptions in the subsequent sections." ("2. Results, page 3, lines 104-107).
Comment 7: The discussion is detailed but sometimes exaggerates conclusions about mechanisms. To be more credible, it is important to present IHC data as ideas to explore, not final answers. This matches scientific standards and encourages more research. Be careful when talking about interactions with other pathways like mTOR and Akt to avoid making broad claims. The unclear parts about cytoplasmic TEAD4 need to be explained better to make the study stronger.
Response 7: Thank you for your valuable comment. In response, we changed our formulations in order to avoid reaching to exaggerating conclusions:
- "However, cytoplasmic and even mitochondrial localization of TEAD4 has been reported in various cell types [26, 27], suggesting broader, possibly context-dependent functions. Mechanistically, cytoplasmic accumulation of TEAD4 in TETs may result from upstream signaling events, such as activation of the MST1–Akt1–mTOR [28, 29] or p38 stress pathways [30]. Alternatively, dominant-negative TEAD4 splice variants, which exhibit dual localization and lack transcriptional activity, might explain the cytoplasmic retention observed in tumors [31]." ("3. Discussion, page 15, lines 377-383).
- "These results point to associations between MST1, LATS1, MOB1A, and TEAD4 expression patterns and tumor stage, which may reflect changes in subcellular dynamics during progression. However, their exact functional contributions remain to be clarified. The observed cytoplasmic shift might be associated with reduced nuclear activity and a tendency toward YAP/TAZ activation in more aggressive tumors" ("3. Discussion, page 16/17, lines 453-458).
- "This observation may point to a shift in Hippo pathway dynamics during tumor progression, yet the regulatory mechanisms underlying these patterns are still undefined and call for mechanistic clarification through future studies" ("3. Discussion, page 17, lines 462-465).
- "Importantly, this is the first study to report cytoplasmic TEAD4 expression—independent of nuclear localization—as a prognostically relevant feature in TETs based on immunohistochemical analysis. While nuclear TEAD4 has previously been linked to prognosis in other malignancies, such as bladder cancer [44, 45], renal clear cell carcinoma [46], and lung adenocarcinoma [47], and one glioma study applied a combined nuclear–cytoplasmic score [48], cytoplasmic TEAD4 has not previously been evaluated as an isolated prognostic variable. These findings may inform future research into non-canonical Hippo signaling and encourage exploration of TEAD4-targeting strategies, especially in tumors characterized by cytoplasmic accumulation. Nevertheless, validation in larger, well-characterized cohorts and functional studies will be necessary to confirm the prognostic and therapeutic relevance of this observation." ("3. Discussion", page 17, lines 480-490).
- "Importantly, the lack of correlation between nuclear YAP1 and TEAD4 expression may indicate a divergence in their regulatory patterns, raising questions about their regulatory role, which should be addressed experimentally" ("3. Discussion", page 18, lines 509-511).
- "Building on these findings, future work should focus on dissecting the underlying biology and assessing clinical applicability" ("3. Discussion", page 18, lines 551-552).
Additionally, we have substantially revised and expanded the discussion of cytoplasmic TEAD4 to provide a clearer and more structured interpretation of its localization and potential significance in TETs. Specifically, we have:
- Clarified the biological relevance of cytoplasmic TEAD4, including a discussion of tumor-specific re-localization, non-transcriptional functions (e.g., involvement in mitochondrial signaling or PI3K/AKT pathways), and potential upstream regulatory influences (e.g., MST1–Akt–mTOR, p38) ("3. Discussion, page 15, lines 376-399.): "[...], which contrasts with TEAD4's canonical role as a nuclear transcription factor [1, 17]. However, cytoplasmic and even mitochondrial localization of TEAD4 has been reported in various cell types [23, 24], suggesting broader, possibly context-dependent functions. Mechanistically, cytoplasmic accumulation of TEAD4 in TETs may result from upstream signaling events, such as activation of the MST1–Akt1–mTOR [25, 26] or p38 stress pathways [27]. Alternatively, dominant-negative TEAD4 splice variants, which exhibit dual localization and lack transcriptional activity, might explain the cytoplasmic retention observed in tumors [28]. In normal thymic tissue, TEAD4 expression was predominantly nuclear, while cytoplasmic staining was only weakly detectable. This indicates that the pronounced cytoplasmic localization observed in TETs likely could represent a tumor-specific alteration rather than a physiological phenomenon. The specificity of the antibody used (Invitrogen PA5-21977, targeting amino acids 1 and 260 of TEAD4) is supported by prior validation in placental tissue, where a similar nuclear and cytoplasmic staining pattern was observed. In our study, lymphocytes remained consistently negative, and staining patterns were reproducible across different cases, further supporting the validity of the observed localization. Taken together, these findings suggest that cytoplasmic TEAD4 in TETs may not merely reflect passive mislocalization, but could point to alternative, non-transcriptional functions or a dysregulation of nuclear transport mechanisms. Whether this cytoplasmic presence represents a functional adaptation contributing to tumorigenesis, or a consequence of disrupted upstream regulation, remains to be determined. Further studies are needed to clarify whether TEAD4 engages in signaling roles outside the nucleus, potentially involving mitochondrial function, oxidative phosphorylation (OXPHOS) [27], or cytoplasmic signaling networks such as PI3K/AKT."
- Included a direct comparison with normal thymic tissue, in which TEAD4 was predominantly nuclear with only weak cytoplasmic staining. This highlights the possible shift as a tumor-associated feature.
- Addressed the specificity and reliability of the observed cytoplasmic signal, including a description of antibody validation and internal controls (e.g., placental tissue, negative lymphocytes), to exclude artefactual staining.
- Separated the mechanistic discussion from the survival analysis, and introduced a dedicated paragraph that focuses on the prognostic relevance of cytoplasmic TEAD4. This avoids redundancy and improves the logical structure of the Results and Discussion sections (see page 17, lines 469-490): "One of the most compelling findings of our study is the identification of a significant association between high cytoplasmic TEAD4 expression and poorer overall survival in patients with TETs. While the majority of Hippo pathway components—including MST1, SAV1, LATS1, MOB1A, TAZ, YAP1, AYAP, and nuclear TEAD4—showed no prognostic impact, the cytoplasmic TEAD4 fraction emerged as a potential prognostic marker. As discussed above, cytoplasmic localization of TEAD4 may result from alternative splicing , disrupted nuclear transport, or upstream signaling alterations, and may reflect either a loss of transcriptional activity or the acquisition of non-nuclear oncogenic functions [25-28]. Although the underlying mechanisms remain speculative, our data suggest that the subcellular redistribution of TEAD4 may not be merely incidental but possibly holding biological and clinical relevance. Importantly, this is the first study to report cytoplasmic TEAD4 expression—independent of nuclear localization—as a prognostically relevant feature in TETs based on immunohistochemical analysis. While nuclear TEAD4 has previously been linked to prognosis in other malignancies, such as bladder cancer [41, 42], renal clear cell carcinoma [43], and lung adenocarcinoma [44], and one glioma study applied a combined nuclear–cytoplasmic score [45], cytoplasmic TEAD4 has not previously been evaluated as an isolated prognostic variable. These findings may inform future research into non-canonical Hippo signaling and encourage exploration of TEAD4-targeting strategies, especially in tumors characterized by cytoplasmic accumulation. Nevertheless, validation in larger, well-characterized cohorts and functional studies will be necessary to confirm the prognostic and therapeutic relevance of this observation."
- In the survival-related section, we now refer back to the mechanistic considerations rather than repeating them, and instead emphasize the novelty of our observation and its potential clinical relevance compared to other tumor entities where nuclear TEAD4 was previously reported as a prognostic factor.
We believe these changes significantly strengthen the manuscript and directly address the reviewer’s concern.
Comment 8: The limitations section should be expanded to talk about issues like not correcting for multiple comparisons, differences in pre-analysis, and limited survival data. This will help people understand the study's limits. Also, avoid making clinical claims without more proof to keep conclusions based on strong evidence. These steps will keep the study honest and make it more influential in the field.
Response 8: Thank you for your comment. We adjusted the limitations section according to your suggestions. All issues are now discussed except for the issue regarding correction for multiple comparisons since it has been performed in our analysis: "First, although the cohort size is reasonable for a rare tumor entity such as TETs, it remains limited and precluded multivariate survival analyses to assess the potential prognostic value of TEAD4 expression while adjusting for relevant clinical and pathological cofactors. Moreover, survival data were only available for a subset of patients, further limiting the statistical power and generalizability of outcome-related findings" ("3. Discussion", page 18, lines 514-519).
Moreover, we tried to avoid making clinical claims by entering the following formulation: "Finally, although the data suggest potential associations with histological subtypes and tumor stage, we refrained from making clinical claims or therapeutic implications due to the exploratory nature of the study and the lack of independent validation"("3. Discussion", page 18, lines 544-547).
Also the differences in pre-analysis were added: "Fourth, potential differences in pre-analytical variables must be considered. Although all tissue samples underwent standardized fixation and storage procedures in an accredited histopathology laboratory, the retrospective study design spanning a period of ten years inherently carries the risk of subtle variations in tissue processing and long-term preservation. For all immunohistochemical analyses, antibody dilutions were selected based on manufacturer recommendations and standardized titration protocols, and staining was performed in a certified and accredited immunohistochemistry laboratory using a consistent detection system throughout. The only deviation from the standard protocol occurred with SAV1, where antigen retrieval conditions were optimized by extending the primary antibody incubation time to nine hours due to initially absent staining. This adjustment resulted in more intense signal, but internal negative tissue areas consistently served as controls. To reduce inter-observer variability, all stainings were scored by the same experienced pathologist. Antibodies that yielded no interpretable or absent staining results under validated conditions were categorically excluded from further analysis" ("3. Discussion", page 18, lines 527-541).
Comment 9: Add scale bars and magnification labels to IHC figures to provide context and enhance credibility. Improve correlation matrix readability through color coding.
Response 9: Thank you for your comment. Scale bars were already included in our figures; however they were not visible since they were covered with the figure labels. In the revised version of the manuscript, we removed the labels in order to make the scale bars visible. Magnification of each figure is also presented in the figure legend, but we added it now also in each figure. A color-coding correlation matrix (heatmap) is used for the table illustrating the correlations between the investigated molecules (Table 2). Positive correlations are now shown in orange. The color intensity indicates the strength of the correlation ("Results", page 14, lines 341-342).
Reviewer 2 Report
Comments and Suggestions for Authors
Dr. Levidou and colleagues present an insightful article exploring the role of Hippo pathway dysregulation in thymic epithelial tumors, with a focus on its association with clinicopathological features and patient prognosis. The study offers valuable contributions to the translational understanding of this signaling pathway. However, several important points should be addressed before the manuscript is suitable for acceptance:
-
Recent studies have demonstrated that YAP/TEAD-mediated Hippo signaling plays a role in resistance to various RTK-driven small molecule inhibitors, thereby reducing their therapeutic efficacy (PMID: 37729426 and PMID: 27105908). The authors should include a brief discussion of this in the introduction to provide a more comprehensive background.
-
Several small-molecule inhibitors targeting the YAP/TEAD interaction are currently under clinical investigation (PMID: 37308716). The authors should mention this ongoing work to highlight the clinical relevance of targeting this pathway.
-
Finally, the authors are encouraged to include their perspective on the translational implications of targeting isoform-specific TEAD versus pan-TEAD inhibition. This would provide useful context regarding therapeutic strategy and drug development.
Author Response
Comment 1: Recent studies have demonstrated that YAP/TEAD-mediated Hippo signaling plays a role in resistance to various RTK-driven small molecule inhibitors, thereby reducing their therapeutic efficacy (PMID: 37729426 and PMID: 27105908). The authors should include a brief discussion of this in the introduction to provide a more comprehensive background.
Response 1: We thank the reviewer for this insightful and constructive comment. In response, we have expanded the introduction to address the broader oncogenic and therapeutic relevance of the Hippo signaling pathway: To address the role of YAP/TEAD signaling in resistance to RTK-targeted therapies, we have added the following sentences to the end of the introductory paragraph on the Hippo pathway: "Beyond its role in tumor progression, YAP/TEAD-mediated signaling has recently been associated with resistance to targeted therapies in other tumor entities, such as KRAS- or EGFR-mutated cancers. There, YAP/TAZ activation enables transcriptional programs that bypass receptor tyrosine kinase (RTK) inhibition, thereby diminishing the efficacy of small-molecule inhibitors [20;21]. These findings emphasize the broader oncogenic relevance of Hippo signaling components and support the investigation of their expression and activity in rare tumors such as TETs." ("1. Introduction, page 3, lines 74-80).
Comment 2: Several small-molecule inhibitors targeting the YAP/TEAD interaction are currently under clinical investigation (PMID: 37308716). The authors should mention this ongoing work to highlight the clinical relevance of targeting this pathway.
Response 2: Thank you for your valuable comment. We agree with that point. To highlight current efforts to therapeutically target YAP/TEAD, we have added the following to the same section: "In parallel, several small-molecule inhibitors designed to disrupt the YAP–TEAD interaction are currently under clinical investigation [22], further underscoring the translational relevance of this pathway." ("1. Introduction, page 3, lines 80-82).
Comment 3: Finally, the authors are encouraged to include their perspective on the translational implications of targeting isoform-specific TEAD versus pan-TEAD inhibition. This would provide useful context regarding therapeutic strategy and drug development.
Response 3: Thank you for your valuable comment. We agree that this discussion is best suited for the "Discussion" section. Accordingly, we have included a brief paragraph reflecting on the potential challenges and opportunities associated with TEAD-targeting strategies in light of our immunohistochemical findings: "The distinct subcellular localization patterns of TEAD4 observed in our study raise important questions regarding the therapeutic modulation of the YAP/TEAD axis. While most current small-molecule inhibitors under clinical investigation aim to broadly disrupt the YAP–TEAD interaction (pan-TEAD inhibition) [22;61], our findings suggest that TEAD4 may exert context-dependent functions that differ between its nuclear and cytoplasmic fractions. In this light, isoform-specific or compartment-targeted strategies could potentially offer a more precise therapeutic approach. TEAD transcription factors are known to exist in multiple isoforms generated through alternative splicing [31], some of which may differ in their transcriptional activity, cofactor binding, or subcellular localization. These isoform-specific properties raise the possibility that broad-spectrum TEAD inhibition could affect both tumor-promoting and non-oncogenic isoforms, potentially limiting therapeutic specificity and increasing off-target effects. Isoform-selective inhibition, on the other hand, may allow for a more refined modulation of TEAD activity—especially in tumors where distinct TEAD isoforms exhibit divergent cellular functions. From a drug development perspective, the challenge lies in designing compounds that selectively inhibit TEAD isoforms or selectively target TEAD activity in particular cellular compartments, such as the cytoplasm or nucleus. Notably, recent studies have demonstrated that allosteric pan-TEAD inhibitors such as GNE-7883 not only suppress YAP/TAZ-mediated transcription but also overcome resistance to targeted therapies—including KRAS-G12C inhibitors—by blocking compensatory YAP/TAZ activation [62]. These findings emphasize the broader applicability of TEAD inhibition in both primary Hippo-dysregulated tumors and resistant tumor states. Whether patients with high cytoplasmic TEAD4 expression might benefit from such therapies remains an important question for future translational research. In summary, these findings underscore the potential relevance of cytoplasmic TEAD4 in TET biology and prognosis and call for further mechanistic and translational investigations to explore its value as a biomarker and therapeutic target." ("3. Discussion, page 19, lines 571-598).
We believe that these additions significantly strengthen the manuscript by enhancing both its scientific depth and translational context.
Round 2
Reviewer 2 Report
Comments and Suggestions for Authors
In line 596, where the authors mention that “studies have demonstrated that allosteric pan-TEAD inhibitors such as GNE-7883 not only suppress YAP/TAZ-mediated transcription but also overcome resistance to targeted therapies—including KRAS-G12C inhibitors—by blocking compensatory YAP/TAZ activation,” they should also include two additional compounds—VT104 (also known as VT3989) and IAG933—which have demonstrated similar effects and are currently in clinical trials (PMID: 37729426 and PMID: 38565920, respectively).
Author Response
Comment 1: In line 596, where the authors mention that “studies have demonstrated that allosteric pan-TEAD inhibitors such as GNE-7883 not only suppress YAP/TAZ-mediated transcription but also overcome resistance to targeted therapies—including KRAS-G12C inhibitors—by blocking compensatory YAP/TAZ activation,” they should also include two additional compounds—VT104 (also known as VT3989) and IAG933—which have demonstrated similar effects and are currently in clinical trials (PMID: 37729426 and PMID: 38565920, respectively).
Response 1: We thank the reviewer for this valuable suggestion. In response, we have revised the relevant paragraph in the discussion to more comprehensively reflect the therapeutic potential of emerging TEAD inhibitors. Specifically, we have added information on the compounds VT104 (VT3989) and IAG933, which are currently in clinical trials and have demonstrated similar efficacy in preclinical models of YAP/TAZ-driven and KRAS-mutant cancers: "Additional compounds currently in clinical development have shown similar capabilities. VT104 (also known as VT3989), an allosteric TEAD inhibitor targeting the TEAD lipid pocket, demonstrated strong preclinical efficacy in models of KRAS-mutant cancers and is currently undergoing clinical evaluation [63]. Furthermore, IAG933, a first-in-class direct disruptor of the YAP–TEAD interface, has shown rapid and potent suppression of TEAD-dependent transcription and robust anti-tumor effects across Hippo-dysregulated and KRAS- or MAPK-altered tumor models—including mesothelioma, NSCLC, and pancreatic ductal adenocarcinoma—and is also in phase I clinical trials [64]." ("3. Discussion, page 19, lines 593-601).